# Deep Semi-Supervised Just-in-Time Learning Based Soft Sensor for Mooney Viscosity Estimation in Industrial Rubber Mixing Process

**DOI:** 10.3390/polym14051018

**Published:** 2022-03-03

**Authors:** Yan Zhang, Huaiping Jin, Haipeng Liu, Biao Yang, Shoulong Dong

**Affiliations:** 1Department of Automation, Faculty of Information Engineering and Automation, Kunming University of Science and Technology, Kunming 650500, China; zhangyanwy126@126.com (Y.Z.); ran@kust.edu.cn (H.L.); biaoykmust@kust.edu.cn (B.Y.); 2Yunnan Key Laboratory of Computer Technologies Application, Kunming 650500, China; 3Department of Chemical Engineering, School of Chemistry and Chemical Engineering, Beijing Institute of Technology, Beijing 100081, China; sldong@bit.edu.cn

**Keywords:** soft sensor, Mooney viscosity, just-in-time learning, semi-supervised learning, stacked autoencoder, gaussian process regression, rubber mixing process

## Abstract

Soft sensor technology has become an effective tool to enable real-time estimations of key quality variables in industrial rubber-mixing processes, which facilitates efficient monitoring and a control of rubber manufacturing. However, it remains a challenging issue to develop high-performance soft sensors due to improper feature selection/extraction and insufficiency of labeled data. Thus, a deep semi-supervised just-in-time learning-based Gaussian process regression (DSSJITGPR) is developed for Mooney viscosity estimation. It integrates just-in-time learning, semi-supervised learning, and deep learning into a unified modeling framework. In the offline stage, the latent feature information behind the historical process data is extracted through a stacked autoencoder. Then, an evolutionary pseudo-labeling estimation approach is applied to extend the labeled modeling database, where high-confidence pseudo-labeled data are obtained by solving an explicit pseudo-labeling optimization problem. In the online stage, when the query sample arrives, a semi-supervised JITGPR model is built from the enlarged modeling database to achieve Mooney viscosity estimation. Compared with traditional Mooney-viscosity soft sensor methods, DSSJITGPR shows significant advantages in extracting latent features and handling label scarcity, thus delivering superior prediction performance. The effectiveness and superiority of DSSJITGPR has been verified through the Mooney viscosity prediction results from an industrial rubber-mixing process.

## 1. Introduction

Rubber mixing is a crucial step in the tire-manufacturing process, where raw materials such as natural rubber or synthetic rubber, additives, and accelerators are mixed together and fed into an internal mixer for processing. After about 2–5 min, a batch of mixing is complete. Mooney viscosity is a key performance indicator that reflects the properties and quality of rubber products [1,2]. However, in actual production process, due to the lack of reliable online measurement equipment, Mooney viscosity can only be obtained through offline analysis in the laboratory, and the sampling period is generally 4–6 h. Such a large measurement delay may not only cause extreme difficulty in grasping the real-time status of the mixing process, but can also lead to economic loss and the waste of raw materials and energy when abnormal operations are found through the measurement of Mooney viscosity. Therefore, the accurate and reliable online measurement of Mooney viscosity is essential for monitoring, controlling and optimizing rubber-mixing production process. To solve this problem, data-driven soft sensor technology has been widely used for online real-time estimation of Mooney viscosity in recent years [3,4,5,6,7]. Such inferential methods realize the real-time and accurate estimation of Mooney viscosity by establishing the mathematical model between the easily measured secondary variables such as the temperature in the mixer cavity, the pressure of the stamping part, the motor speed and the motor power and the primary variable Mooney viscosity.

Until now, data-driven soft sensor methods for industrial Mooney viscosity prediction mainly include partial least squares (PLS) [8], Gaussian process regression (GPR) [3], extreme learning machine (ELM) [4] and deep learning (DL) [7]. However, these methods are essentially global modeling techniques, which cannot effectively deal with strong nonlinearity and multi-mode characteristics. This is because global methods rely on a single regression model, and strive to obtain good generalization performance in all process regions. However, due to the complex characteristics of the rubber-mixing process, global soft sensor models may not be able to accurately describe some local-region characteristics, thus leading to poor prediction performance. Therefore, Mooney viscosity prediction methods based on local modeling techniques have attracted the most attention over the past several years [3,4,5].

Just-in-time learning (JIT) [3,7] is one of the most commonly used local modeling techniques in soft sensor applications. It follows the philosophy of “divide-and-conquer” and builds a localized prediction model to obtain accurate prediction results. JIT modeling is based on the basic assumption that “similar inputs produce similar outputs”, and when a prediction task arrives, only a small number of similar samples are selected from the historical database to construct a local prediction model instead of all of the modeling data. This method can not only deal with the complex process characteristics effectively, but can also avoid the high computational burden caused by using all the modeling data. Therefore, in recent years, various JIT soft sensor methods have been proposed for Mooney viscosity prediction in industrial rubber-mixing processes. For example, Jin et al. (2017) [4] proposed a just-in-time regularized extreme learning machine (JRELM) soft sensor method, which showed better results in the prediction of Mooney viscosity in industrial rubber mixing compared to traditional methods. Jin et al. (2019) [9] proposed a soft sensor modeling framework EMO-EJIT, based on diverse weighted Euclidean distance similarity and optimized the similarity parameters by evolutionary multi-objective optimization method. Jin et al. (2020) [3] proposed an ensemble just-in-time learning Gaussian-process regression soft sensor modeling method based on multimodal perturbation, referred to as MP-EJITGPR. However, in practical applications, many JIT soft sensor modeling methods for Mooney viscosity estimation still encounter poor performance due to improper feature selection or extraction and insufficient labeled data.

As is widely recognized, it is very important to select appropriate input data features for constructing accurate soft sensors. Doubtlessly, this holds true for JIT soft sensor development for Mooney viscosity estimation. In general, feature-selection methods can be divided into two categories: direct selection and latent feature extraction. The former selects some features directly from the original features through some ranking criterion or an optimization approach, such as filtering, wrapping, and embedded selection, etc., while the latter maps the original features to a new feature space through a linear or nonlinear transformation, with principal component analysis as a typical representative of this kind of methods. However, there are still two difficulties in selecting appropriate features to construct accurate soft sensor models, namely (1) the direct selection methods may lead to certain information loss due to discarding some variables; and (2) traditional latent feature extraction methods cannot effectively mine complex process data features in industrial processes.

In recent years, as an emerging machine-learning technology, deep learning techniques [10] have been extensively applied in the fields of machine learning, natural-language processing and image recognition. By introducing multi-level nonlinear mapping, this type of method can extract abstract features from complex process data and effectively deal with complex industrial-process data modeling, and thus has been widely used in soft sensor applications [11,12,13,14,15]. As one of the most commonly used deep learning methods, stacked autoencoder (SAE) has been widely used for soft sensor development. Yuan et al. (2018) [16] proposed a new variable-wise weighted stacked autoencoder (VM-SAE) for extracting latent features from industrial data. For example, Yuan et al. (2020) [17] proposed a semi-supervised stacked autoencoder (SS-SAE) and applied it to two industrial cases, and the results show that the proposed method is superior to the traditional deep-learning soft sensor methods. Sun and Ge (2020) [18] designed a gated stacked target-related autoencoder (GSTAE) to improve the prediction performance of soft sensor.

Moreover, traditional Mooney viscosity soft sensor methods mainly focus on a supervised learning framework, which requires sufficient labeled training samples to ensure good prediction performance. However, in an actual rubber-mixing production process, the labeled data that can be used for Mooney viscosity soft sensor modeling are seriously insufficient due to the limitations of technical or economic conditions such as high cost and long period of Mooney viscosity offline analysis. In practice, besides the difficult-to-measure variable, a large number of easily measured process data have been preserved thanks to the rapid development and wide application of a distributed control system and online measurement technology. For this reason, the phenomenon of “unlabeled data rich but labeled data poor” is very common in the industrial rubber-mixing production process. However, traditional supervised soft sensor methods often ignore the exploitation of unlabeled data, since the unlabeled data, whose Mooney viscosity measurements are lacking, also contain valuable information about the process. Thus, semi-supervised learning has been recently introduced to leverage both labeled and unlabeled data, aiming at developing high-performance semi-supervised soft sensors [11,19].

Therefore, methods to effectively use semi-supervised learning to improve Mooney viscosity prediction performance has become a research hotspot [5,6,7]. For example, Zheng et al. (2018) [5] proposed a semi-supervised extreme learning machine soft sensor method (JSELM), which integrates just-in-time learning, extreme learning machine and graph Laplacian regularization into a unified online-modeling framework. It can improve the performance of a soft sensor by mining information in a large number of unlabeled data. Zheng et al. (2017) [6] proposed a semi-supervised Mooney viscosity prediction method by combining an extreme learning machine with graph Laplacian regularization and by introducing a bagging ensemble strategy. Zheng et al. (2020) [7] proposed an ensemble deep correntropy kernel regression (EDCKR) robust semi-supervised soft sensor method, which integrates ensemble learning, deep brief network, and correntropy kernel regression into a unified framework.

In general, semi-supervised learning can be divided into the following five categories: generative models [20], self-training [21], co-training [22], graph-based methods [23], semi-supervised support vector machines (S3VM) [24]. Additionally, the combination of semi-supervised learning and other methods has received increasing attention in recent years, such as the combination of semi-supervised learning and ensemble learning [9,19,25,26,27], and the combination of semi-supervised learning and deep learning [28,29,30,31]. According to the use of unlabeled data, semi-supervised soft sensors are roughly divided into two categories, namely pseudo-label estimation and unlabeled data regularization. Among them, the main representatives of the former are self-training and co-training semi-supervised soft sensors, while the latter are represented by generative models, graph-based methods, and semi-supervised deep learning methods. In the context of JIT soft sensor modeling, the lack of labeled samples is a key factor restricting the performance of JIT learning soft sensor for Mooney viscosity estimation. A realistic solution to this problem is to expand the training sample set by obtaining high-confidence pseudo-labeled data. Moreover, these methods have many advantages, such as being less affected by model assumptions and non-convexity of loss functions, simple and effective implementation, and the employment of wrapping modeling principles. Thus, we focus on pseudo-label estimation-based semi-supervised soft sensor modeling for Mooney viscosity estimation in this study.

The core issue of the semi-supervised learning soft sensor based on pseudo-label estimation is how to obtain high-confidence pseudo-labeled data. In the field of classification, two typical pseudo-label estimation methods, i.e., self-training and co-training, have been widely applied. Self-training is a kind of semi-supervised learning method in which the learner predicts the unlabeled data, obtains the pseudo-labeled data, and iteratively adds high-confidence pseudo-labeled data to expand the labeled training set, so as to guide its continuous learning. However, this kind of method only uses a single learner, and is prone to fall into local optimum and leads to the inaccurate estimation of pseudo labels. Different from this, the co-training framework improves model prediction by building different learners on multiple views, and then each learner iteratively feeds the other learners with high-confidence pseudo-labeled data to expand the training set, so as to improve the prediction performance of the model. However, it is still difficult to obtain high-confidence pseudo-labeled data in regression tasks because it is more likely to cause error accumulation and propagation due to iterative learning for traditional pseudo-label estimation methods, and ultimately leads to poor model performance.

In light of the above problems, a novel deep semi-supervised just-in-time Gaussian process regression (DSSJITGPR) soft sensor method is proposed for Mooney viscosity estimation in industrial rubber-mixing processes. This method can effectively combine the advantages of JIT learning, semi-supervised learning and deep learning, thus providing a new way to achieve accurate predictions of Mooney viscosity. The main contributions of this work are summarized as follows:A stacked autoencoder-based deep learning technique is used to extract the latent feature information from the process data of industrial rubber mixing, which is superior to traditional feature selection and extraction methods in handling high-dimensional, complex process data.An evolutionary pseudo-labeling optimization approach is proposed to expand the modeling database with limited labeled data by obtaining high-confidence pseudo-labeled data. The basic idea of this approach is to first formulate an explicit pseudo-labeling optimization problem and then solve this problem using an evolutionary approach.By integrating JIT learning, semi-supervised learning and deep learning into an online modeling framework, DSSJITGPR can provide much better prediction accuracy than traditional soft sensors for Mooney viscosity estimation in industrial rubber-mixing process.

The rest of the paper proceeds as follows. Section 2 provides a brief introduction of JIT learning, SAE, and GPR. The proposed DSSJITGPR soft sensor method is described in Section 3. Section 4 reports a case study to demonstrate the feasibility and efficiency of the proposed approach. Finally, conclusions are drawn in Section 5.

## 2. Preliminaries

### 2.1. Just-in-Time Learning

Just-in-time Learning (JIT) [3], also known as lazy learning [32] or locally weighted learning [33], etc., is a popular local learning framework for nonlinear system modeling. Based on the principle of “similar input produces similar output” and the “divide-and-conquer” philosophy, JIT methods aim to construct local predictive models to obtain accurate prediction results. The basic principle of JIT learning is illustrated in Figure 1. Different from traditional global modeling methods, it has the following features: (1) All input and output data are stored in a database; (2) Only those samples most relevant to the query sample are selected for modeling according to a certain similarity measure; and (3) The already built local model is discarded after completing the prediction task.

When a new query sample $x_{new}$ arrives, the online implementation steps for JIT learning are as follows: (1) Select historical data similar to the query sample $x_{new}$ from the database according to a certain similarity criterion; (2) Build the local model based on the selected similar historical samples; (3) Complete the prediction of primary variables and then discard the model. Repeat the above steps when a new query point arrives.

As can be seen from Figure 1, the definition of a similarity measure is the key to constructing high-performance JIT soft sensor models. The most commonly used similarity criteria include distance-based similarity functions, such as Euclidean distance [34,35] and Mahalanobis distance [36], angle-based similarity functions, such as cosine similarity [37,38], and similarity functions based on correlation coefficients, such as Pearson correlation coefficient similarity [39]. Among them, Euclidean-distance similarity is the simplest and most commonly used similarity criterion for JIT soft sensor modeling. It evaluates the point-to-point linear distance between two sample points in space. However, the Euclidean distance ignores the differences between input variables, so various weighted distance similarity criteria are proposed, among which covariance weighted-distance (CWD) similarity [9,40,41,42,43] is one representative example. CWD considers both the relationships between input variables and between input and output variables. Thus, in this work, a CWD similarity measure is chosen for JIT soft sensor modeling, and its definition is as follows:(1)$$\begin{array}{c} \omega_{i}=\exp\left(-\frac{d_{i}}{\sigma_{d}\varphi }\right) \end{array}$$
(2)$$\begin{array}{c} d_{i}=\sqrt{{\left(x_{i}-x_{new}\right)}^{T}H\left(x_{i}-x_{new}\right)} \end{array}$$
(3)$$\begin{array}{c} H=\frac{{\left(X^{T}y\right)}^{T}\left(X^{T}y\right)}{{\left\|X^{T}y\right\|}^{2}} \end{array}$$
where $\sigma_{d}$ is the standard deviation of $d_{i}\left(i=1,2,\cdots ,N\right)$, φ is a localized parameter, H represents the weighted matrix, X and y represent the input and output matrices, respectively.

### 2.2. Stacked Autoencoder

A stacked autoencoder (SAE) [44] is a deep neural network model which is stacked by multiple autoencoders (AEs) [45] layer by layer. It has gained great success in a wide range of applications such as feature extraction, image recognition, fault diagnosis, data reduction and denoising. The structure of an autoencoder is shown in Figure 2. It mainly consists of three layers of network, namely the input layer, hidden layer and output layer. The input and hidden layers form the encoding network of AE, while the hidden and output layers form the decoding network of AE. For a given input vector $x={\left[x_{\left(1\right)},x_{\left(2\right)},\ldots ,x_{\left(d_{x}\right)}\right]}^{T}$, $d_{x}$ is the input dimension. The encoding and decoding process of AE can be described as follows:(4)$$\begin{array}{c} h=f\left(Wx+b\right) \end{array}$$
(5)$$\begin{array}{c} \tilde{x}=g\left(\tilde{W}h+\tilde{b}\right) \end{array}$$
where $h={\left[h_{\left(1\right)},h_{\left(2\right)},\ldots ,h_{\left(d_{h}\right)}\right]}^{T}$ is the hidden layer input, $d_{h}$ is the hidden variable dimension, $\tilde{x}={\left[{\tilde{x}}_{\left(1\right)},{\tilde{x}}_{\left(2\right)},\ldots {\tilde{x}}_{\left(d_{x}\right)}\right]}^{T}$ is the outputs of the autoencoder, i.e., the reconstruction inputs; f(∙) and g(∙) are the activation functions, which generally take the nonlinear activation function Sigmoid; W is the weight matrix between the input layer and the hidden layer neurons, and b is the bias between the input layer and the hidden layer neurons; $\tilde{W}$ is the weight matrix between the output layer and the hidden layer neurons; and $\tilde{b}$ is the bias between the output layer and the hidden layer neurons.

The training of an autoencoder can be achieved by iteratively optimizing the parameter set $\Theta =\left\{W,\tilde{W},b,\tilde{b}\right\}$ using a back propagation (BP) algorithm and gradient descent algorithm, which aims to minimize the loss function in terms of the reconstruction error, and thus learns the latent representation of features in the hidden layer. The reconstruction error function and parameter update formula can be expressed as follows:(6)$$\begin{array}{c} J\left(\Theta \right)=\frac{1}{2N}\sum_{i=1}^{N}\left(\|{\tilde{x}}_{i}-x_{i}\|{}^{2}\right) \end{array}$$
(7)$$\begin{array}{c} \{\begin{array}{c} W=W-\alpha \frac{\partial J\left(\Theta \right)}{\partial W}\\ b=b-\alpha \frac{\partial J\left(\Theta \right)}{\partial b} \end{array} \end{array}$$
where N is the total number of the training samples.

SAE is composed of multiple AEs stacked layer by layer, where each layer of AE is trained separately, thus facilitating better processing capabilities of complex abstract data features. Specifically, the outputs of the former AE hidden layer are directly used as the inputs of the latter AE hidden layer, and so on to obtain an SAE network with different number of hidden layers. The network structure of SAE is shown in Figure 3.

The training of the SAE network can be divided into two stages: unsupervised greedy pre-training and supervised fine-tuning, as shown in Figure 4. In the pre-training stage, each AE is trained layer by layer, and the hidden layer outputs of the previous AE are used as the hidden layer inputs of the next AE, and this process is repeated until all AEs are initialized. In the fine-tuning stage, an output layer is connected to the output end of the SAE network, and the back propagation algorithm and gradient descent algorithm are used to fine-tune the parameters of the SAE network to achieve a better data dimension reduction effect.

### 2.3. Gaussian Process Regression

Gaussian process (GP) is a set of random variables, any finite number of which obey joint Gaussian distribution [46,47]. Consider a dataset $Z=\left\{X,y\right\}={\left\{x_{i},y_{i}\right\}}_{i=1}^{n}$, its regression model can be described as follows:(8)$$\begin{array}{c} y=f\left(x\right)+\varepsilon ,\varepsilon \sim N\left(0,\sigma_{n}^{2}\right) \end{array}$$
where x is the input data, n is the number of input samples, f(∙) is the unknown regression function, and ε is Gaussian noise with a mean value of 0 and variance of $\sigma_{n}^{2}$. From the perspective of function space, the Gaussian process can be completely determined by covariance function $C\left(x,x^{\prime }\right)$ and mean function m(x), which are defined as follows:(9)$$\begin{array}{c} \{\begin{array}{ccc} m\left(x\right) & =E\left[f\left(x\right)\right] & \\ C\left(x,x^{\prime }\right) & =E[(f\left(x\right)-m\left(x\right))(f\left(x^{\prime }\right)-m\left(x^{\prime }\right))] \end{array} \end{array}$$

Therefore, the Gaussian process can be described as:(10)$$\begin{array}{c} f\left(x\right)\sim \mathrm{GP}\left(m\left(x\right),C\left(x,x^{\prime }\right)\right) \end{array}$$

In general, the modeling data is preprocessed by normalization, and we can assume that the training sample set is generated from a zero-mean Gaussian process:(11)$$\begin{array}{c} y\sim \mathrm{GP}\left(0,C\right) \end{array}$$
where, C is a symmetric positive definite covariance matrix of order n×n, the elements of which can be expressed as $C_{ij}=C\left(x_{i},x_{j}\right)$.

When a query sample $x_{new}$ arrives, the joint prior distribution of training output y and test output $y_{new}$ can be expressed as:(12)$$\begin{array}{c} \left[\begin{array}{c} y\\ y_{new} \end{array}\right]\sim N\left(0,\left[\begin{array}{cc} C & k_{new}\\ k_{new}^{T} & C\left(x_{new},x_{new}\right) \end{array}\right]\right) \end{array}$$
where $k_{new}={\left[C\left(x_{new},x_{1}\right),\ldots ,C\left(x_{new},x_{n}\right)\right]}^{T}$ is the n×1 order covariance matrix between the query sample $x_{new}$ and the input x, $C\left(x_{new},x_{new}\right)$ represents the covariance of $x_{new}$ itself. Since the posterior distribution of $y_{new}$ satisfies $y_{new}|X,y,x_{new}\sim N\left({\hat{y}}_{new},\sigma_{new}^{2}\right)$, the predicted mean ${\hat{y}}_{new}$ and variance $\sigma_{new}^{2}$ of test sample $x_{new}$ can be calculated as follows:(13)$$\begin{array}{c} \{\begin{array}{c} {\hat{y}}_{new}=k_{new}^{T}C^{-1}y\\ \sigma_{new}^{2}=C\left(x_{new},x_{new}\right)-k_{new}^{T}C^{-1}k_{new}^{T} \end{array} \end{array}$$

Compared with other modeling methods, Gaussian process regression (GPR) not only has strong nonlinear-processing ability, but can also provide the prediction uncertainty.

Covariance function is a key component of GPR modeling, which encodes the a priori assumption of the function. In the Gaussian process, the covariance function defines the proximity or similarity of data points. In this paper, the Matérn covariance function with the noise term is chosen and its definition is as follows:(14)$$\begin{array}{c} C\left(x_{i},x_{j}\right)=\sigma_{f}^{2}\left(1+\frac{\sqrt{3}\|x_{i}-x_{j}\|}{l}\right)\exp\left(-\frac{\sqrt{3}\|x_{i}-x_{j}\|}{l}\right)+\sigma_{n}^{2}\delta_{ij} \end{array}$$
where $\Theta =\left\{\sigma_{f}^{2},l,\sigma_{n}^{2}\right\}$ is the set of hyperparameters, $\sigma_{f}^{2}$ is the output scale, l is the input scale, and $\sigma_{n}^{2}$ is the noise variance. $\delta_{ij}$ is 1 when i=j, and 0 otherwise.

In addition, the hyperparameter set Θ can be determined by maximizing the logarithmic likelihood function of historical data:(15)$$\begin{array}{c} \underset{\Theta }{\max}\log p\left(y|X\right)=-\frac{1}{2}y^{T}C^{-1}y-\frac{1}{2}\log\left|C\right|-\frac{n}{2}\log\left(2\pi \right) \end{array}$$

Usually, the conjugate gradient method, Newton’s method and other methods can be used to solve the above optimization problem. In such cases, partial derivatives of hyperparameters need to be obtained:(16)$$\begin{array}{c} \frac{\partial \left[\log p\left(y|X\right)\right]}{\partial \Theta }=-\frac{1}{2}\mathrm{tr}\left(C^{-1}\frac{\partial C}{\partial \Theta }\right)+\frac{1}{2}y^{T}C^{-1}\frac{\partial C}{\partial \Theta }C^{-1}y \end{array}$$
where tr(∙) represents the trace of the matrix.

## 3. Proposed DSSJITGPR Soft Sensor Method for Mooney Viscosity Estimation

In traditional soft sensor methods for Mooney viscosity estimation, input variables are usually selected using some relevance evaluation criterion or extracted through projection mapping methods. However, such feature selection or extraction strategies cannot always function well for characterizing complex process characteristics, especially for high-dimensional process data. Moreover, JIT methods often encounter the scarcity of labeled data and thus result in poor prediction performance. Thus, we attempt to develop enhanced JIT soft sensors for Mooney viscosity estimation by exploring the following two channels: (1) applying deep learning techniques to extract latent features behind the industrial rubber mixing process data; and (2) augmenting the labeled modeling database through an evolutionary pseudo-labeling optimization. The details of the proposed DSSJITGPR approach will be described in the following subsections.

### 3.1. Latent Feature Extraction

To ensure the prediction accuracy of Mooney viscosity soft sensors, it is of great importance to select or extract informative features relevant to the output variable. Since Mooney viscosity is an end-use quality variable in industrial rubber mixing process, all process data during the batch running can be potentially used as the model inputs, thus leading to a high-dimensional modeling data set. Thus, a popular deep learning technique, i.e., SAE, is adopted to achieve latent feature extraction due to its strong capability of extracting the abstract features from complex process data.

The principle of feature extraction using SAE method is shown in Figure 5, where $\left\{x_{1},x_{2},\ldots ,x_{d}\right\}$ denotes the original input features, i.e., temperature, pressure, power, motor speed, energy, etc., {k,k−1,k−2,…,k−m} are the selected time instants in a batch of run, k is the prediction time of endpoint Mooney viscosity, and $\left\{h_{1},h_{2},\ldots ,h_{p}\right\}$ represents the latent features obtained from SAE feature extraction.

To obtain reliable latent features, SAE training is achieved based on the mixing data by merging the limited labeled data with abundant unlabeled data, where all available input data are used for pre-training while the labeled data are used for fine tuning.

### 3.2. Acquisition of High-Confidence Pseudo-Labeled Data

Since it is too time-consuming and expensive to obtain sufficient Mooney viscosity measurements through a laboratory analysis, the lack of labeled data has become a key factor in restricting the performance of JIT soft sensors. Thus, we attempt to enhance the JIT soft sensor performance by leveraging both labeled and unlabeled data. One efficient solution to this problem is to build semi-supervised soft sensors based on the pseudo-labeling approach, which aims to expand the labeled training set by obtaining high-confidence pseudo-labeled data, as illustrated in Figure 6. In such cases, the method used to obtain high-confidence pseudo-label data is the core of semi-supervised learning based on pseudo-label estimation.

Traditionally, self-training and co-training are two frequently used methods for pseudo-label estimation [21,22]. Self-training obtains the pseudo-labeled data through iterative learning, where a supervised learner is first built from only the labeled data and then refined by adding high-confidence pseudo-labeled data. Differently, the co-training paradigm constructs different learners on multiple diverse views, and then each learner iteratively provides the other learners with high-confidence pseudo-labeled data to expand the training set, so as to improve the prediction performance of the model. Based on self-training and co-training, a variety of semi-supervised methods have been proposed, including Tri-training [48], Co-Forest [49], Co-Training by Committee (CoBC) [50], Multi-train [51], etc. Although many of these methods have achieved good results in classification tasks, it is still difficult to obtain high-confidence pseudo-labeled data in regression tasks. As an early attempt, Zhou and Li (2007) [52] extended co-training to regression applications for the first time, and proposed the Co-regression method COREG. However, these semi-supervised methods largely depend on the iterative learning process, which is prone to cause error accumulation and propagation and thus lead to poor model performance. To address this issue, in a recent study [19], we proposed a novel way of obtaining high-confidence pseudo-labeled data through evolutionary optimization and preliminarily verified its effectiveness.

Thus, in this work, we aim to augment the labeled database by applying the pseudo-labeling optimization approach. The main idea of this method is to transform the pseudo-label estimation problem into an explicit optimization problem and solve it through an evolutionary optimization approach.

#### 3.2.1. Formulation of Pseudo-Labeling Optimization Problem

Given a labeled data set $L={\left\{\left(x_{l,i},y_{l,i}\right)\right\}}_{i=1}^{n_{l}}$ and an unlabeled data set $U={\left\{x_{u,i}\right\}}_{i=1}^{n_{u}}$, where $x_{l}$ and $y_{l}$ represent the real input and output values, $x_{u}$ is the unlabeled sample, $n_{l}$ and $n_{u}$ represent the numbers of labeled and unlabeled samples, respectively; and $Y_{u}={\left\{y_{u,i}\right\}}_{i=1}^{n_{u}}$ is the pseudo-label set corresponding to the unlabeled dataset U, and the true labels are unknown. Let $U^{\prime }$ denote a subset with $n_{u}^{\prime }$ samples randomly drawn from the unlabeled data set U, for which $Y_{u}^{\prime }={\left\{y_{u,i}^{\prime }\right\}}_{i=1}^{n_{u}^{\prime }}$ is the corresponding pseudo-label set. Then, an explicit optimization problem should be defined to estimate $Y_{u}^{\prime }$ using the pseudo-labeling optimization approach.

Naturally, we can assume that the modeling data for Mooney viscosity estimation are independent and identically distributed, i.e., L and $U^{\prime }$ are drawn from the same process distribution. Thus, ideally, the unknown functional relationships described by labeled data and pseudo-labeled data should be consistent. Let $h_{l}$ be the prediction model built from the labeled data L, $h_{u}$ be the prediction model built from the pseudo-labeled set $L_{u}=\left\{U^{\prime },Y_{u}^{\prime }\right\}$, and $h_{l+u}$ be the prediction model learnt with the extended labeled data $L_{l+u}=L\cup L_{u}$ after including the pseudo-labeled data. Following the aforementioned assumption, the prediction error of model $h_{u}$ on training set L should be minimized, i.e.,
(17)$$\begin{array}{c} y_{u}^{*}=\arg\underset{y_{u}}{\min}\frac{1}{n_{l}}\sum_{i=1}^{n_{l}}{\left(y_{l,i}-{\hat{y}}_{l,i}\right)}^{2} \end{array}$$
where $y_{l}$ is the real output of input $x_{l}$, ${\hat{y}}_{l}$ is the predicted output of model $h_{u}$ for $x_{l}$, and $y_{u}=\left[y_{u,1},y_{u,2},\ldots y_{u,n_{u}^{\prime }}\right]$ represents the decision variables.

As can be seen, Equation (17) represents the prediction accuracy of the model $h_{u}$ on the labeled data. In addition, another optimization objective can be defined based on the prediction error of $h_{l+u}$ on the labeled training set after adding the pseudo-labeled data. That is, the following objective should be minimized:(18)$$\begin{array}{c} y_{u}^{*}=arg\underset{y_{u}}{\min}\frac{1}{n_{l}}\sum_{i=1}^{n_{l}}{\left(y_{l,i}-{\hat{y}}_{l+u,i}\right)}^{2} \end{array}$$
where ${\hat{y}}_{l+u,i}$ is the predicted output of model $h_{l+u}$ for $x_{l}$.

By combining the two objectives in Equations (17) and (18), the final objective function for estimating the pseudo-labels ${\left\{y_{u,i}^{\prime }\right\}}_{i=1}^{n_{u}^{\prime }}$ can be given as
(19)$$y_{u}^{*}=\arg\underset{y_{u}}{\min}\frac{1}{n_{l}}\overset{n_{l}}{\underset{i=1}{\sum }}{\left(y_{l,i}-{\hat{y}}_{l,i}\right)}^{2}+\lambda \frac{1}{n_{l}}\overset{n_{l}}{\underset{i=1}{\sum }}{\left(y_{l,i}-{\hat{y}}_{l+u,i}\right)}^{2}\;s.t.y_{min,i}\leq y_{u,i}\leq y_{max,i}\;,\;i=1,2,\ldots ,n_{u}^{\prime }$$
where 0<λ≤1 is a controlling parameter for balancing two types of objectives.

#### 3.2.2. Solving of Pseudo-Labeling Optimization Problem

The aim of this step is to solve the optimization problem shown in Equation (19). Since classic optimization approaches such as analytical and numerical methods usually have high requirements of the continuity and differentiability of the objective function, they are ill-suited for solving the formulated pseudo-labeling optimization problem. Alternatively, evolutionary optimization [53] methods are essentially global optimization approaches and require no specific requirements on the objective function. They exhibit the characteristics of self-organization, self-learning and self-adaptation, and are not limited by the nature of the problem.

As one of the most typical evolutionary algorithms, the genetic algorithm (GA) [54] searches for the optimal solution by simulating the biological evolution process of natural selection and genetic mechanisms. Since it can effectively manage complex and nonlinear optimization problems, GA has been widely used in feature selection, signal processing, machine learning. Therefore, GA is used to solve the pseudo-labeling optimization problem described in Section 3.2.

A schematic diagram of obtaining high-confidence pseudo-labeled data through evolutionary optimization is presented in Figure 7. The main steps are as follows:
Select an unlabeled subset $U^{\prime }$ from U and the pseudo-labels $Y_{u}^{\prime }={\left\{y_{u,i}^{\prime }\right\}}_{i=1}^{n_{u}^{\prime }}$ serve as decision variables.Using real-number coding, $Y_{u}^{\prime }$ is coded as a chromosome, as shown in Figure 8.An initial population with $n_{pop}$ individuals is randomly generated within the ranges of $y_{min,i}\leq y_{u,i}\leq y_{max,i},\;i=1,2,\ldots ,n_{u}^{\prime }$.Evaluate the fitness of each individual in the population according to the reciprocal of the objection function values.Generate a new population by performing selection, crossover and mutation operations, and return to step (4).If the stopping condition for GA optimization is satisfied, the optimal solution with the highest fitness is selected and decoded as the pseudo-label estimation of the unlabeled sample data $y_{u}^{*}=\left[y_{u,1}^{*},y_{u,2}^{*},\ldots ,y_{u,n_{u}^{\prime }}^{*}\right]$. Consequently, a pseudo-labeled data set can be obtained.Merge the labeled and pseudo-labeled data to form an enlarged labeled data set; subsequently, an enhanced GPR model $h_{l+u}$ is built.Evaluate the performance enhancement ration (*PER*) $h_{l+u}$ over $h_{l}$ on the validation set $L_{val}=\left\{X_{val},y_{val}\right\},$ that is,
(20)$$\begin{array}{c} PER=\frac{RMSE_{init}-RMSE_{enhanced}}{RMSE_{init}} \end{array}$$
where $RMSE_{enhanced}$ denotes the root mean squared error (*RMSE*) of $h_{l+u}$ on $L_{val}$ while $RMSE_{init}$ is the prediction RMSE from $h_{l}$. If the model performance enhancement has arrived at the presetting threshold, add the obtained pseudo-labeled data to the database,, otherwise discard these data.

Repeat the above steps until the stopping condition has been achieved.

To assure the pseudo-labeling optimization performance, the optimal parameters $\Phi =\left\{n_{u}^{\prime },\lambda \right\}$ for the proposed pseudo-label estimation method are determined according to the validation set:(21)$$\begin{array}{c} \Phi^{*}=\arg\underset{\Phi }{\min}{\left\|y_{val}-{\hat{y}}_{val}\right\|}^{2} \end{array}$$
where ${\hat{y}}_{val}$ is the predicted output of $X_{val}$ using the proposed semi-supervised soft sensor model.

In addition, the calculation of fitness involves the repeated reconstruction of the GPR model, thus resulting in a relatively large computational burden. Hence, a parameter-sharing strategy is employed to reduce the computational complexity of GPR modeling. The basic idea of this strategy is to save the hyperparameters obtained by model $h_{l}$ training, and then these hyperparameters are directly used in the subsequent optimization processes to train GPR models based on pseudo-labeled data. This strategy is based on the assumption that labeled and unlabeled data are independent and identically distributed, so the hyperparameters of the GPR model trained by them should also be very similar. In addition, the parameter sharing strategy can also mitigate the negative effects of repeatedly reconstructing the GPR model with unreal information during the optimization process.

It should also be noted that, in addition to GA algorithm, the above-mentioned pseudo-labeling optimization problem can also be solved by using other evolutionary optimization methods, such as differential evolution (DE) and particle swarm optimization (PSO).

### 3.3. Implementation Procedure

The overall implementation procedure of the proposed DSSJITGPR soft sensor method is shown in Figure 9. At the offline phase, latent features are extracted from the unlabeled and labeled data based on SAE, then the modeling database is augmented by including high-confidence pseudo-labeled data, which are obtained through an evolutionary pseudo-labeling optimization approach. At the online prediction phase, a semi-supervised JITGPR model is built for Mooney viscosity estimation based on the modeling data in latent space, CWD similarity measure, and the augmented database. Then, the established model is discarded. When a new query sample arrives, the online modeling and prediction steps are repeated.

## 4. Application to an Industrial Rubber Mixing Process

The effectiveness and superiority of the proposed DSSJITGPR soft sensor method have been verified through the Mooney viscosity prediction of an industrial rubber mixing process. The methods for comparison are as follows:PLS: the global PLS model.GPR: the global GPR model.ELM [55]: the global ELM model.SSELM [56]: the global semi-supervised ELM model;CoGPR: the co-training based GPR model using two sets of randomly selected input variables as different views.JITGPR: the JIT learning based GPR model using CWD similarity measure.DPLS: the deep learning based PLS model using SAE for latent feature extraction.DGPR: the deep learning based GPR model using SAE for latent feature extraction.DCoGPR: the deep learning based CoGPR model using SAE for latent feature extraction.DJITGPR: the deep learning based JITGPR model using SAE for latent feature extraction.DSSGPR: the deep learning based semi-supervised GPR model using SAE for latent feature extraction and including the pseudo-labeled data to the labeled training set.DSSJITGPR (the proposed method): the deep semi-supervised JITGPR model.

The abovementioned methods can roughly be classified into the following four groups: (1) global supervised learning models, i.e., PLS, GPR, ELM, DPLS, and DGPR; (2) local supervised learning models, i.e., JITGPR and DJITGPR; (3) global semi-supervised learning models, i.e., SSELM, CoGPR, DCoGPR and DSSGPR; and (4) local semi-supervised learning model, i.e., DSSJITGPR, the method proposed in this paper.

To evaluate the prediction performance of different soft sensor models, the root mean square error (RMSE) and coefficient of determination ($R^{2}$) are used:(22)$$\begin{array}{c} RMSE=\sqrt{\frac{1}{N_{\mathrm{test}}}\overset{N_{\mathrm{test}}}{\underset{i=1}{\sum }}{\left({\hat{y}}_{i}-y_{i}\right)}^{2}} \end{array}$$
(23)$$\begin{array}{c} R^{2}=1-\frac{\sum_{i=1}^{N_{\mathrm{test}}}{\left({\hat{y}}_{i}-y_{i}\right)}^{2}}{\sum_{i=1}^{N_{\mathrm{test}}}{\left({\hat{y}}_{i}-\bar{y}\right)}^{2}} \end{array}$$
where $N_{\mathrm{test}}$ represents the number of testing samples; ${\hat{y}}_{i}$ and $y_{i}$ represent the predicted and actual values of output variable, respectively, and $\bar{y}$ is the mean value of the actual output variable.

### 4.1. Process Description

Rubber is a key material mainly used in vehicle tires, as well as a vast array of other articles ranging from conveyor belts to examination gloves. Rubber mixing is an important procedure in rubber processing and production, which aims to mix the natural rubber or synthetic rubber, additives, accelerators and other raw materials according to a certain process formula. During the rubber-mixing process, Mooney viscosity is an important indicator for monitoring the rubber product quality.

Mooney viscosity is usually measured by rotating a standard rotor in the sample of the airtight chamber under certain conditions. The shear resistance of the rotor rotation is related to the viscosity change of the sample in the vulcanization process, which can be displayed on the dial with Mooney as the unit through the dynamometer. This shear resistance torque is defined as Mooney viscosity. The higher the Mooney viscosity is, the higher the molecular weight is, the wider the distribution range is, the lower the plasticity is, and it becomes difficult to mix evenly and perform extrusion processing. It also affects the fluidity of the compound at the initial stage of vulcanization, and quality problems become common, such as unclear edges and corners of the molding pattern. On the contrary, the lower the molecular weight, the narrower the distribution range, the greater the plasticity, the harder to mix, and the lower the tensile strength after vulcanization. In short, Mooney viscosity represents the molecular weight, reflecting the processing performance of rubber—mainly the fluidity of rubber. To obtain good product performance and semi-finished product stiffness, the Mooney viscosity should be large; whereas to guarantee the easy processing of semi-finished products and adhesive penetration, the Mooney viscosity should be small. Therefore, the Mooney viscosity has been accepted as an important quality variable to indicate the rubber quality in industrial rubber-mixing processes, the real-time measurements of which are crucial for performing advanced monitoring, controlling and optimization of rubber mixing processes.

However, in industrial rubber production, it usually takes 4–6 h to obtain the Mooney viscosity offline analysis value after the completion of each batch of the mixing process. Compared to the batch mixing duration of 2–5 min, such a large measurement delay may not only cause extreme difficulty in grasping the real-time status of the mixing process but also lead to economic loss and a waste of raw materials and energy when abnormal operations are found through the measurement of Mooney viscosity. Fortunately, soft sensor techniques enable the real-time accurate estimations of Mooney viscosity. In certain application scenarios, hardware sensors can be replaced with high-performance soft sensors, or offline laboratory analysis frequency can be reduced significantly due to the use of soft sensors, which significantly reduces the investment costs with regard to purchasing equipment. Overall, accurate and reliable online measurements of Mooney viscosity based on soft sensors are highly valuable for achieving effective monitoring, control, and optimization of rubber mixing production processes. Thus, we attempt to develop soft sensors to achieve real-time and accurate estimations of Mooney viscosity to ensure the optimum stability of rubber-mixing processes.

The industrial rubber-mixing process under study is conducted in a tire-production enterprise in East China. The production site and process flow diagram are shown in Figure 10 and Figure 11, respectively. Since a complete batch of rubber mixing process corresponds to only one end-point Mooney viscosity measurement, the process variables corresponding to time 0 s, 14 s, 18 s, 22 s, ..., 118 s are used as input variables for soft sensor development, including temperature in the mixer chamber, motor power, stamping pressure, motor speed and energy, etc.

### 4.2. Prediction Results and Analysis

The labeled and unlabeled modeling data are collected from the distributed control system and laboratory analysis, and some obvious outliers in input and output data are eliminated by a simple 3σ rule. As a result, a total of 1172 batches of process data were collected, including 730 unlabeled data and 442 labeled data. The labeled data is further divided into the training set (150 samples), test set (172 samples) and validation set (120 samples).

In order to obtain high prediction performance, the following key parameters for different soft sensor methods are determined by minimizing the prediction performance on the validation set:
The numbers of principal components for PLS and DPLS are nine and 5five respectively.The number of hidden layer neurons in ELM is 455.The number of hidden layer neurons in SSELM is 170, and the weight coefficient of Laplacian regularization is 0.6.The iteration for CoGPR and DCoGPR is 70, and the number of high-confidence pseudo-labeled selected for each iteration is five.The number of local modeling samples for JITGPR, DJITGPR and DSSJITGPR is *L* =15.In our proposed DSSJITGPR method, the parameters of GA optimization for pseudo-label estimation are set to $n_{pop}=30$, $n_{gen}=30$. In addition, the optimal parameter combination of $\left\{n_{u}^{\prime },\lambda \right\}$ is selected as {120,0.7}.With the SAE network structure determined as 140-70-30-5-1, the detailed parameter settings are listed in Table 1.

In addition, in this paper, a parameter-sharing strategy was used to improve the modeling efficiency, mainly aiming to reduce the computational complexity of GPR modeling during GA optimization. When other parameters used in the DSSJITGPR method remain unchanged, use of the parameter-sharing strategy leads to a run time of 19 s to complete GA optimization, while the time required to complete a run of GA optimization is about 696 s when the parameter-sharing strategy is not used.

Table 2 compares the prediction results from different soft sensor methods. Some findings can be summarized as follows:
Linear model-based soft sensors, i.e., PLS and DPLS, are obviously inferior to nonlinear methods due to the failure in dealing with nonlinear characteristics of the process.Despite the use of nonlinear modeling techniques, the prediction performance of traditional global modeling methods such as GPR, DGPR, and ELM is still very poor. Compared with global modeling, the local-learning soft sensors such as JITGPR and DJITGPR have achieved a certain degree of prediction performance improvement.In general, whether global or local, or supervised or semi-supervised, the prediction performance of different soft sensors was greatly improved after introducing SAE base feature extraction. These results reveal the necessity and effectiveness of SAE-based feature extraction for the high-dimensional process data.In most cases, the introduction of semi-supervised learning is helpful for enhancing the prediction accuracy of supervised soft sensors. However, in this case study, SSELM does not achieve significant performance enhancement, which is mainly because of improper unlabeled data introduction. In contrast, compared with GPR and DGPR, the prediction performance of Co-GPR and DCoGPR becomes worse due to the improper construction of different views or unreliable pseudo-label estimation. Compared with DGPR, the performance of DSSGPR was improved due to the use of the expanded database, indicating that the high-confidence pseudo-labeled data obtained using the proposed pseudo-labeling optimization approach are reliable. In addition, when local learning is introduced, DSSJITGPR provides much better prediction results than DSSGPR.


The above results show that, due to the efficient combination of SAE-based feature extraction, evolutionary pseudo-labeling optimization-based data augmentation, and JIT learning-based local modeling, DSSJITGPR provides accurate Mooney viscosity predictions that are significantly superior to traditional global/local and supervised/semi-supervised soft sensors for Mooney viscosity prediction.

In addition, Figure 12 shows the scatter plot of the actual and predicted Mooney viscosity values obtained by different GPR based soft sensor methods. The closer the scatter is to a diagonal, the higher the prediction accuracy of this method is. It can be seen that the scatter points shown in Figure 12a,c and e are far from the diagonal line, indicating that GPR, CoGPR and JITGPR provide poor prediction accuracy. In contrast, the remaining scattered points are very close to the diagonal line, which implies that the prediction performance of DGPR, DCoGPR, DJITGPR, DSSGPR and DSSJITGPR provide much better prediction performance. Overall, the proposed DSSJITGPR approach provides the best prediction results.

In order to intuitively evaluate the prediction performance of DSSJITGPR, Figure 13 shows the trend plots of Mooney viscosity prediction results using this method. It can be seen that the predicted values of Mooney viscosity are highly consistent with the actual values, which further verifies the effectiveness of the proposed soft sensor method. Interestingly, in industrial rubber mixing process, in order to meet the market demands, different raw material formulations are usually used to produce rubber compounds of different specifications, resulting in multi-mode characteristics of the process data and large differences in Mooney viscosity. In this paper, the process data under two formulations are mainly studied, that is, two operation modes are involved in our case study. As a result, as illustrated in Figure 13, the viscosity values accumulate around the two values.

Moreover, Figure 14 compares the prediction performance of JITGPR, DJITGPR and DSSJITGPR under different local modeling sizes. It is easy to see from the figure that the DSSJITGPR model, which combines deep learning and semi-supervised learning, performs better than the JITGPR and DJITGPR models. In addition, from the perspective of prediction robustness, compared with the JITGPR soft sensor method, the performance curves of DJITGPR and DSSJITGPR under different local modeling sizes are more stable, which verifies that they are much less sensitive to local modeling size. Once again, these results show that the proposed DSSJITGPR method is more accurate and reliable than traditional JIT soft sensors in predicting Mooney viscosity.

## 5. Conclusions

In this paper, a deep semi-supervised just-in-time learning soft sensor modeling method, referred to as DSSJITGPR, was proposed for Mooney viscosity prediction in the industrial rubber-mixing process. The main contributions of the proposed method are summarized as follows: (1) By applying a popular deep learning technique, namely a stacked autoencoder, the latent features are extracted from the historical process data, which effectively exploits the latent structure and reduces input data dimension. (2) The estimation problem of pseudo labels is formulated as an optimization problem and solved through evolutionary optimization, thus obtaining high-confidence pseudo-labeled data to expand the labeled modeling database. (3) With GPR as the base learning technique, an enhanced semi-supervised JITGPR model can be built for online quality prediction with JIT technology, so as to effectively manage the complex characteristics in the industrial rubber-mixing process, while avoiding the problem of high computational burden caused when using all the Mooney viscosity data. By combining the merits of JIT learning, semi-supervised learning, and deep learning paradigms, the proposed DSSJITGPR method significantly outperforms traditional soft sensor methods for the Mooney viscosity prediction task. In the follow-up work, two issues deserve further study: (1) How to obtain high-confidence pseudo-labeled data by other methods? (2) How to introduce ensemble learning into the proposed framework to further improve the prediction performance of Mooney viscosity?

## Figures and Tables

**Figure 1 polymers-14-01018-f001:**
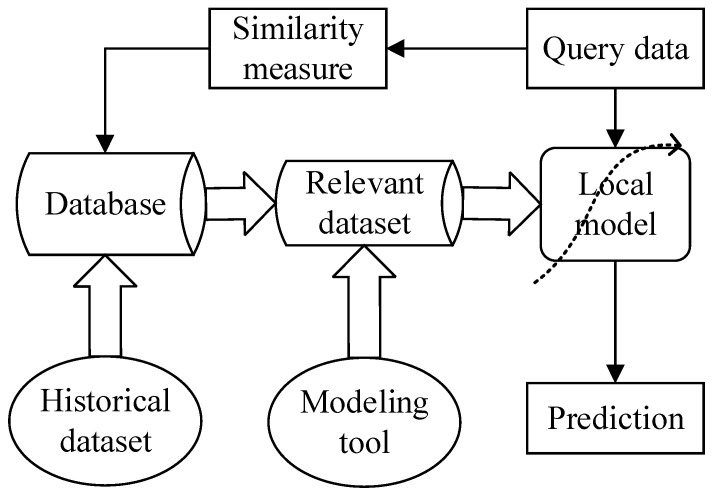
Just-in-time learning framework.

**Figure 2 polymers-14-01018-f002:**
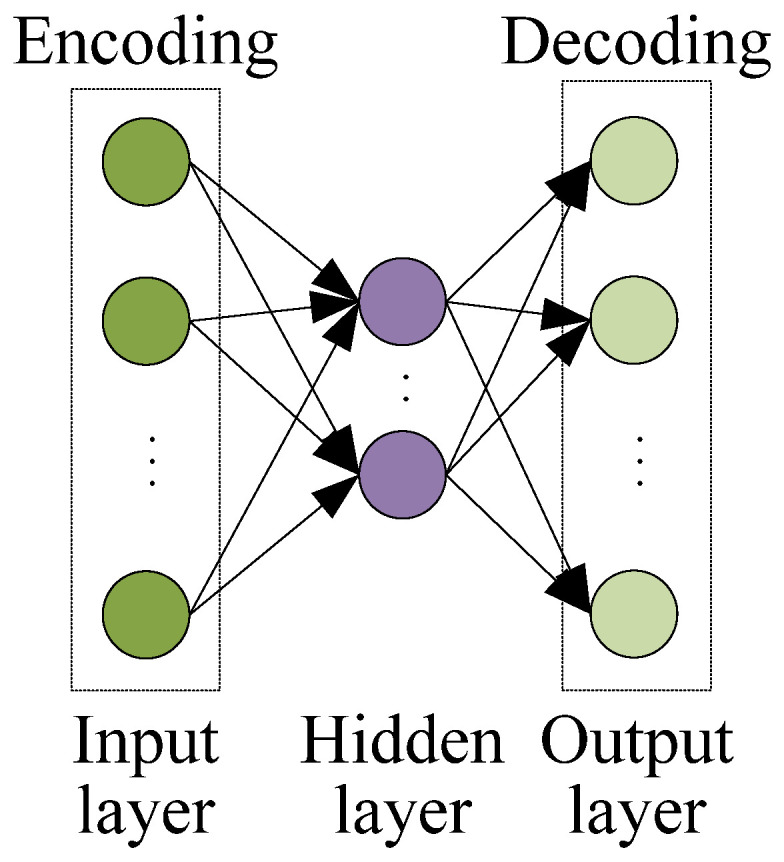
Structure of an autoencoder.

**Figure 3 polymers-14-01018-f003:**
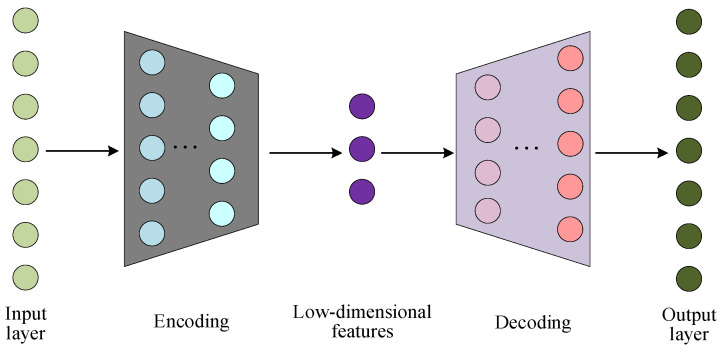
Structure of a stacked autoencoder.

**Figure 4 polymers-14-01018-f004:**
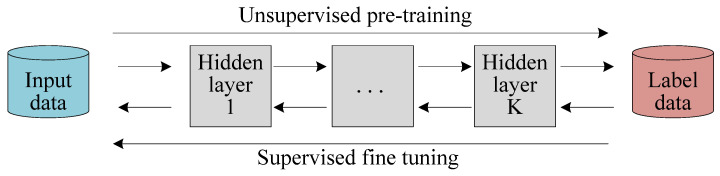
Training process of a SAE network.

**Figure 5 polymers-14-01018-f005:**
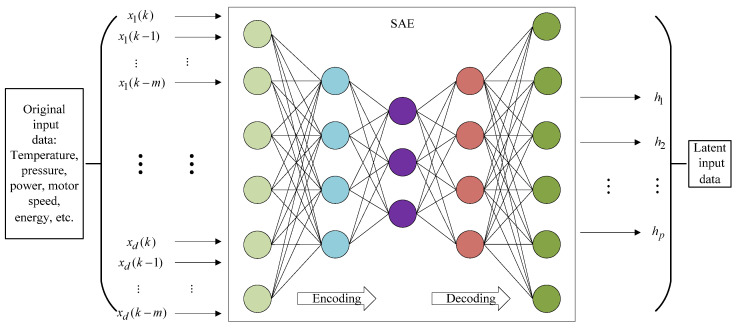
SAE-based latent feature extraction for Mooney viscosity prediction.

**Figure 6 polymers-14-01018-f006:**
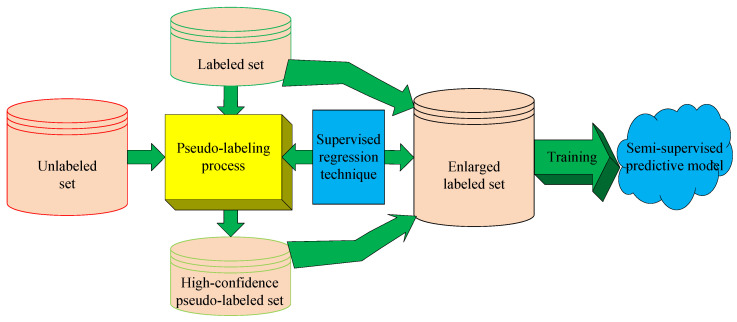
Basic principle of semi-supervised soft sensor modeling based on pseudo-label estimation.

**Figure 7 polymers-14-01018-f007:**
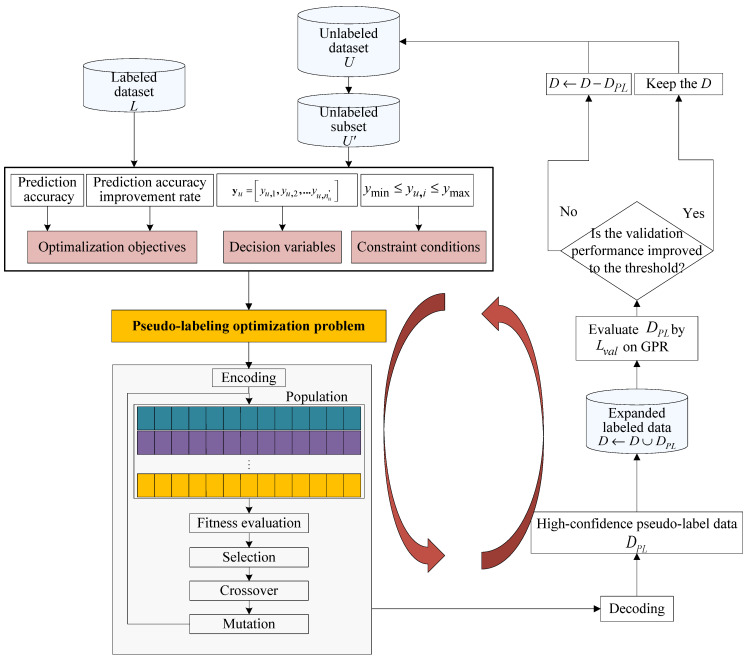
Schematic diagram for obtaining high-confidence pseudo-labeled data through evolutionary optimization.

**Figure 8 polymers-14-01018-f008:**
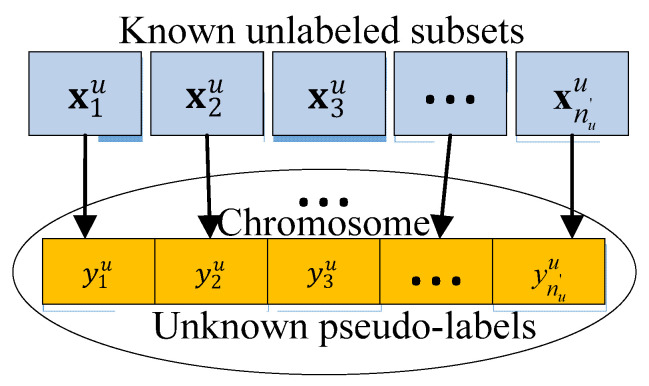
Individual structure for GA based pseudo-labeling optimization.

**Figure 9 polymers-14-01018-f009:**
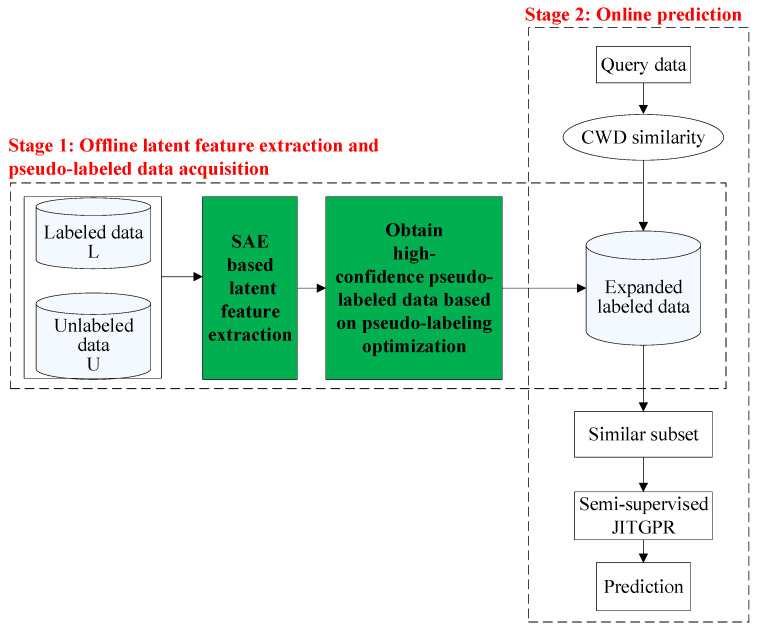
Schematic diagram of implementing the proposed DSSJITGPR soft sensor.

**Figure 10 polymers-14-01018-f010:**
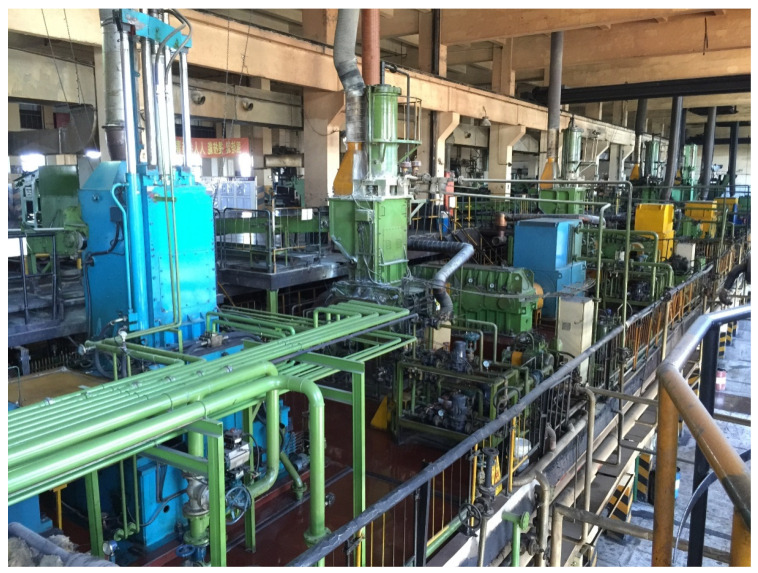
Production workshop of industrial rubber-mixing process.

**Figure 11 polymers-14-01018-f011:**
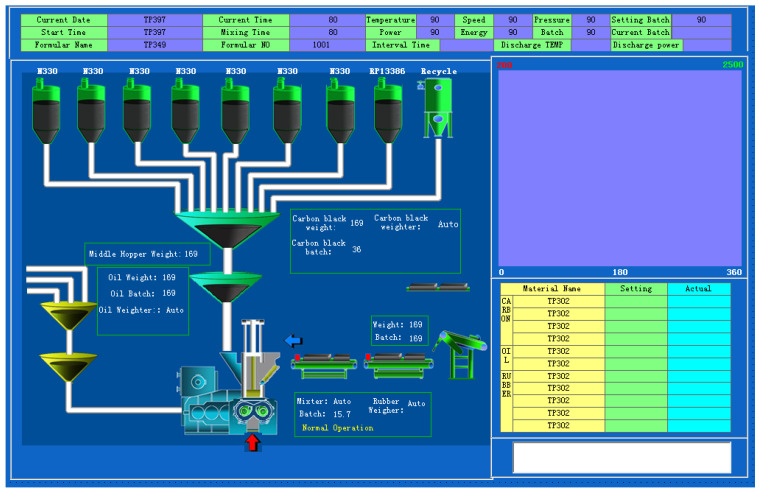
Process flow diagram.

**Figure 12 polymers-14-01018-f012:**
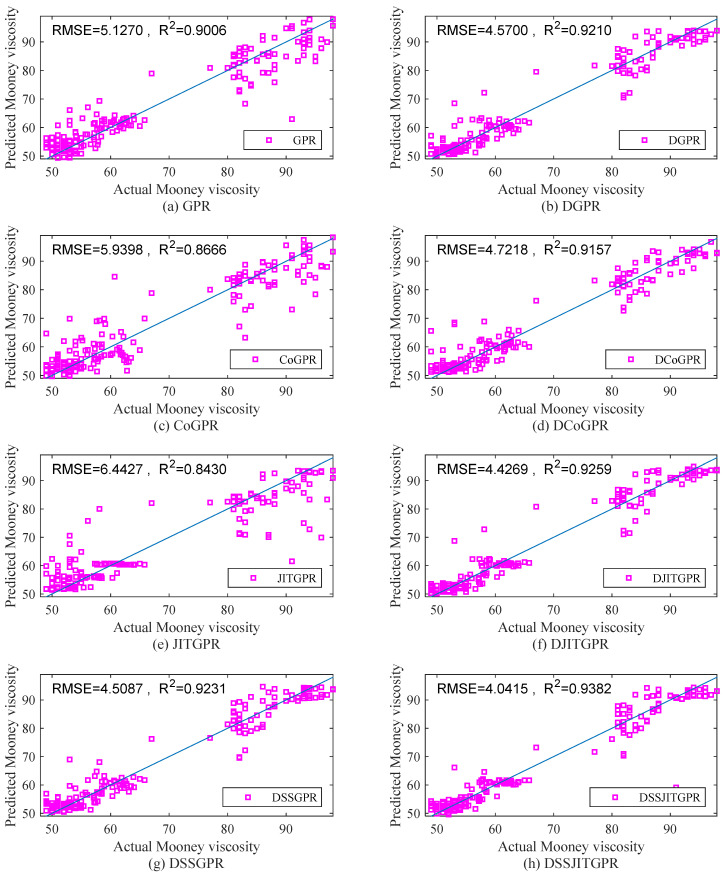
Scatter plots of prediction results using different soft sensor methods. (**a**) GPR; (**b**) DGPR; (**c**) CoGPR; (**d**) DCoGPR; (**e**) JITGPR; (**f**) DJITGPR; (**g**) DSSGPR; (**h**) DSSJITGPR.

**Figure 13 polymers-14-01018-f013:**
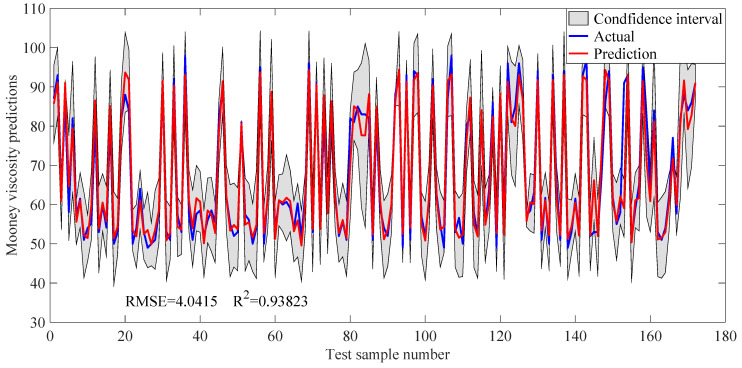
Trend plots of Mooney viscosity predictions using the proposed DSSJITGPR approach (*L* = 15).

**Figure 14 polymers-14-01018-f014:**
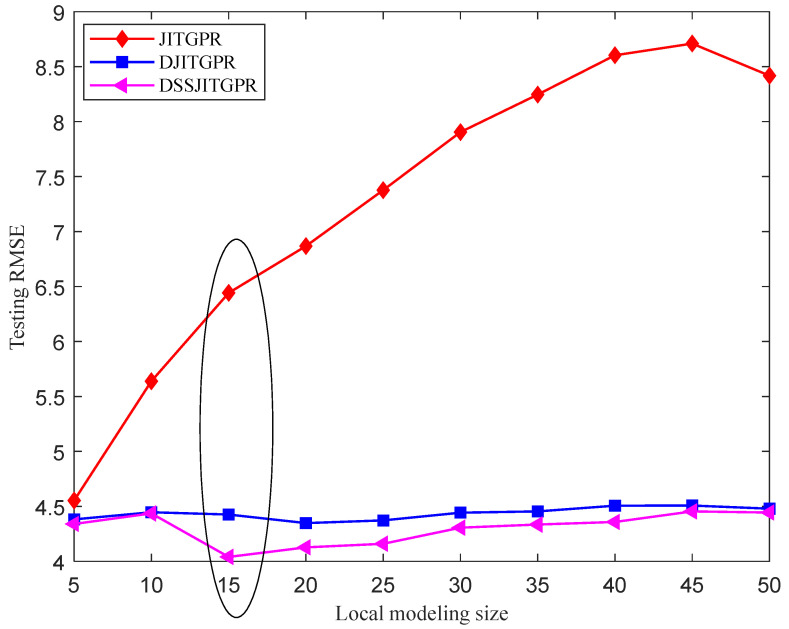
Prediction performance of three soft sensor methods under different local modeling sizes.

**Table 1 polymers-14-01018-t001:** Parameter settings for SAE network structure and training.

Symbol	Description	Value
*Hnode1*	Number of nodes in the first layer	70
*Hnode2*	Number of nodes in the second layer	30
*Hnode3*	Number of nodes in the third layer	5
*Lrate1*	Pre-training learning rate	0.05
*Nepoch1*	Epoch number in pre-training	300
*Bsize1*	Sample batch size in pre-training	20
*Lrate2*	Fine-tuning learning rate	0.07
*Nepoch2*	Epoch number in fine-tuning	300
*Bsize2*	Sample batch size in fine-tuning	20

**Table 2 polymers-14-01018-t002:** Comparison of Mooney viscosity prediction results using different soft sensors (*L* = 15).

No.	Method	RMSE	$R^{2}$
1	PLS	7.4703	0.7889
2	GPR	5.1270	0.9006
3	ELM	6.7405	0.8279
4	SSELM	6.6534	0.8319
5	CoGPR	5.9398	0.8666
6	JITGPR	6.4427	0.8430
7	DPLS	5.3602	0.8913
8	DGPR	4.5700	0.9210
9	DCoGPR	4.7218	0.9157
10	DJITGPR	4.4269	0.9259
11	DSSGPR	4.5087	0.9231
12	DSSJITGPR	**4.0415**	**0.9382**

## Data Availability

The data presented in this study are available on request from the corresponding author.

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
