# Peer review of "Deep Semi-Supervised Just-in-Time Learning Based Soft Sensor for Mooney Viscosity Estimation in Industrial Rubber Mixing Process"

_polymers, 2022, doi:10.3390/polym14051018_

Round 1

Reviewer 1 Report

I really have enjoyed reading the manuscript. It consists of well designed feature extraction and semi-supervised learning steps, which are essential in soft sensor design. I have a number of suggestions and questions:

  1. The authors use covariance weighted distance for feature selection. Are these features the original process variables, or the features offered by the encoder? I believe the former, since they use JITGPR during online prediction phase, so why not use the features from the autocoder for the query samples as well? Furthermore, after using all these nonlinear tools, is it a good idea to use the global covariance function to determine the "importance" of features in JITL? Have they tried using the simple kNN in JITL?
  2. Since you are using weighted Eucledian distances, and this is a topic, which, I believe, is not sufficiently covered in the literature, could you please also cite these two related papers?

- S. Kim, R. Okajima, M. Kano, S. Hasebe, Development of soft-sensor using locally weighted PLS with adaptive similarity measure, Chemometr. Intell. Lab. Syst. 124 (2013) 43–49.

- B. Alakent, Online tuning of predictor weights for relevant data selection in just-in-time-learning, Chemometr. Intell. Lab. Syst. 203 (2020) 104043,

  1. What is the rationale of using the optimization objective function in Eqn. 17, while there is already Eqn. 18? Why does GPR need to use only the unlabeled data (hu) to predict the labeled ones? In reality, the labeled data exists, hence it is sufficient that the combination of labeled and unlabaled data should give good results. Indeed, the authors have determined the weight of Eqn. 18 (l) equal to 0.7 via cross validation, showing that deviation from unity may be due to the small sampling size.
  2. How long does the optimization take in real time?
  3. Are the batches divided into training, validation and test sets randomly, or in time sequence? If the latter one is used, could you show these intervals on Fig. 13? Additionally, the legend in Fig. 13 seems to be incorrect; confidence intervals should be the shaded region.
  4. What is the total number of features (variables ´ number of time lags)?
  5. Viscosity values seem to be accumalted around two values. Could you very briefly comment on this bimodal process behavior?
  6. A number of typos:

Line 147 on page 3, "Moreover, this" should be corrected to "Moreover, these"

Line 570 on page 17, ‘Liner” should be corrected to “Linear”

Please use a superscript in typing the square in “R2”.

Reviewer 2 Report

The revised article is of interest for those who work in the rubber sector for tires, production and recycling, but there are aspects that must be improved and clarified:

In conclusions justify because the Deep learning DSSJTGPR improves the soft sensor Methods for the prediction of Mooney Viscosity, reason and justify this statement in the Conclusions section.

In Figure 14 I do not understand the x, y variables in the graph, develop the meaning of the variables, and why it highlights the point x = 15?

References must be correlative, and this does not happen to the introduction

Why Mooney Viscosity is so important and what parameters do you have? Are not real measurements of the Mooney Viscosity instead of the predictive techniques that the article relates? I think real measures should be carried out so I do not understand the need to predict the value of Mooney Viscosity, this point is very important to clarify it

What is 3σ rule? (Line 546)

In Figure 13, which is L = 15?

Round 2

Reviewer 1 Report

The authors have done the required changes in the manuscript and responded satisfactorily to my concerns. I do not have further revision requests.

Author Response

Thank you again for your time and energy to improve our paper.

Reviewer 2 Report

The questions from reviewer has been answered satisfactorily, but the paper reflect few changes from the last version, please introduce some aspects from your questions to final paper and highlight in the delivered final version

Author Response

Thank you again for your time and energy to improve our paper. Thank you very much for this valuable question. In response to your comments and suggestions, in the latest manuscript, we added some text descriptions and marked them with the "Track Changes" function of MS Word. Please see the latest “polymers-1595932” to find them.